# Thyroid Hormone Signaling Is Required for Dynamic Variation in Opsins in the Retina during Metamorphosis of the Japanese Flounder (*Paralichthys olivaceus*)

**DOI:** 10.3390/biology12030397

**Published:** 2023-03-02

**Authors:** Yaxin Shi, Yang Shi, Wenyao Ji, Xike Li, Zhiyi Shi, Jilun Hou, Wenjuan Li, Yuanshuai Fu

**Affiliations:** 1Key Laboratory of Freshwater Aquatic Genetic Resources, Ministry of Agriculture, Shanghai Ocean University, Shanghai 201306, China; 2Key Laboratory of Exploration and Utilization of Aquatic Genetic Resources, Ministry of Education, Shanghai Ocean University, Shanghai 201306, China; 3Shanghai Collaborative Innovation for Aquatic Animal Genetics and Breeding, Shanghai Ocean University, Shanghai 201306, China; 4Shanghai Food Research Institute, Shanghai 200235, China; 5Hebei Key Laboratory of the Bohai Sea Fish Germplasm Resources Conservation and Utilization, Beidaihe Central Experiment Station, Chinese Academy of Fishery Sciences, Qinhuangdao 066100, China

**Keywords:** *Paralichthys olivaceus*, opsin, thyroid hormone, metamorphosis

## Abstract

**Simple Summary:**

Fish adapt to changes in their external light environment by adjusting their visual system. Fish that live at different depths can perceive different spectral ranges, leading to differences in the type and expression of their optic proteins. The Japanese flounder (*Paralichthys olivaceus*) is a valuable cold temperate benthic marine fish that undergoes a series of metamorphoses during its juvenile stage. The vision of flounder larvae is reshaped to adapt to changes in their light environment. Previous studies have shown that thyroid hormone (TH) signaling is critical for flounder metamorphosis. In the present study, we investigated the tissue composition and the metamorphosis temporal expression profile of five opsin genes in flounder larvae. We also investigated the effect of TH on the opsin genes’ expression levels by adding TH and thiourea (a TH inhibitor, TU) to the culture water. In addition, we investigated the thyroid hormone receptor (TR)-mediated targeting regulatory relationship between TH and the opsin genes to further explore the function of TH in visual remodeling during flounder metamorphosis.

**Abstract:**

In the present study, we investigated the function of thyroid hormones (TH) in visual remodeling during Japanese flounder (*Paralichthys olivaceus*) metamorphosis through cellular molecular biology experiments. Our results showed that the expression of the five opsin genes of the flounder were highest in eye tissue and varied with the metamorphosis process. The expression of *rh*1, *sws*2*aβ* and *lws* was positively regulated by exogenous TH, but inhibited by thiourea (TU) compared to the control group. In addition, there was a significant increase in *sws*2*aβ* and *lws* in the rescue experiments performed with TU-treated larvae (*p <* 0.05). Meanwhile, T3 levels in flounder larvae were increased by TH and decreased by TU. Based on the differences in the expression of the three isoforms of the thyroid hormone receptor (TR) (Trαa, Trαb and Trβ), we further hypothesized that T3 may directly or indirectly regulate the expression of *sws*2*aβ* through Trαa. This study demonstrates the regulatory role of TH in opsins during flounder metamorphosis and provides a basis for further investigation on the molecular mechanisms underlying the development of the retinal photoreceptor system in flounders.

## 1. Introduction

Fish can sense and adapt to the light environment of their habitat through their vision, and different photoreceptor (PR) subtypes within the retina determine their level of vision and ability of color vision. Most fish contain the following two PR types: rod photoreceptors and cone photoreceptors. The rod synthesizes rhodopsin (Rh1), which is responsible for vision in low light, and the cone synthesizes red-sensitive opsin (M/Lws), green-sensitive opsin (Rh2), blue-sensitive opsin (Sws2) and ultraviolet-sensitive opsin (Sws1), which have color vision functions [1,2,3,4]. Opsins belong to the category of G-protein-linked membrane receptors (GPCRs) [5], and play a very important role in transmitting extracellular signals into the cell interior [6]. Visual systems of fish are unusually complex because their opsins have remarkable diversity [7,8]. Flatfish are a special kind of marine fish with eyes on the same side of their body, and a diversity of opsin genes, which is very different from other fish. There are ten opsin genes in Japanese flounders (*Paralichthys olivaceus*) [9], nine in turbot (*Scophthalmus maximus*) [10], eight in barfin flounders (*Verasper moseri*) [11], five in tongue sole (*Cynoglossus semilaevis*) [12], five in winter flounders (*Pseudopleuronectes americanus*) [13] and five in Atlantic halibut (*Hippoglossus hippoglossus*) [14].

Japanese flounders are an important marine-culture fish in China. As is the case with other flatfish, Japanese flounders also undergo drastic metamorphosis from larva to the juvenile stage [15,16], characterized by a change in body shape from symmetrical to asymmetrical, migration of the right eye towards the left, as well as a transition from a pelagic to benthic habitat, and thus from light to dim photic environments [17]. At present, ten opsin genes (*rh*1, *lws*, *sws*1, *sws*2*aα*, *sws*2*aβ*, *sws*2*b, rh*2*a-*1, *rh*2*a-*2, *rh*2*b-*1 and *rh*2*b-*2) have been identified in Japanese flounders [9]. However, the molecular mechanism of visual adaptation to changes in photic environments during metamorphosis is still unknown.

The thyroid hormone (TH) accelerates metamorphosis and shortens the transition time from pelagic to benthic habitats in Japanese flounders, while thiourea (TU) slows metamorphosis by inhibiting the synthesis of TH [18]. TH plays a key role in many physiological functions, and the active form of action is triiodothyronine (T3) [19,20,21]. T3 functions by binding to the thyroid hormone receptor (TR), a ligand-dependent nuclear transcription factor that binds to specific TH response elements (TRE) in the promoter region of target genes, thereby regulating gene transcription [22,23]. TH and TR are essential for the differentiation of photoreceptors and the development of the retina [24,25,26,27,28]. In mice, TH signaling can promote the expression of M-cone opsin and suppress the expression of S-cone opsin through TR and play a positive regulatory role in promoting the patterned distribution of dorsal–ventral opsins [29,30,31,32]. However, the molecular mechanism of TH signaling that regulates visual adaptation to changes in photic environments during metamorphosis is still unknown.

In this study, we demonstrate visual remodeling during flounder metamorphosis by constructing spatiotemporal expression patterns of opsin genes, analyzing the influence of exogenous TH on the expression levels of opsins during flounder metamorphosis to explain the important role of TH in regulating visual remodeling, and preliminarily revealing the molecular mechanism of TH that regulates visual remodeling by establishing the regulatory relationship of TH, TR and opsins.

## 2. Materials and Methods

### 2.1. Animal Experiment

The flounder samples and tissues were collected from the Beidaihe Central Experiment Station of the Chinese Academy of Fishery Sciences (Qinhuangdao city, China). The flounders were intensively cultured in seawater at 19–21 °C and a salinity of 33 for 4 days after hatching. We fed the larvae with nutrition-fortified rotifers (*Brachionus plicatilis*) and *Artemia nauplii*. Larvae at 16 days post-hatching (dph) were randomly divided into 3 groups (three replicates per group). The normal control group (NC) was cultured with normal seawater; the TH group was exposed to seawater with 0.1 mg/L exogenous TH (T3; Sangon, Shanghai, China) over the entire period of the experiment; and the TU group was cultured in seawater containing 30.0 mg/L exogenous thiourea (TU; Sangon, Shanghai, China) [33]. 

### 2.2. Sample Collection

All experimental flounders were anesthetized with MS-222, euthanized by decapitation, and rinsed with DEPC water before sample collection. Following the schedule described by Minami [34], larvae samples (*n* = 3) were periodically collected at 17 dph, 20 dph, 24 dph, 28 dph, 32 dph, 36 dph, and 41 dph. After microscopic observation, metamorphosis stage larval specimens, including the normal control, TH-treated and TU-treated larvae, were collected. Adult flounders were dissected separately for tissues containing the brain, eye, gill, heart, liver, stomach, kidney, intestine, gonads and muscle (*n* = 3). All samples were rapidly frozen in liquid nitrogen and subsequently stored in −80 °C refrigerator. 

For the rescue group sample collection, at 36 dph, TU-treated larvae were divided into the following three groups (three replicates per group): a TU group that continued to be cultured in seawater containing 30.0 mg/L TU; a TU + NC group cultured in natural seawater; and a TU + TH group cultured in seawater containing 0.1 mg/L TH. The three groups were further cultured until 41dph, and then collected as samples and stored at −80 °C for subsequent experiments. 

### 2.3. Phylogenetic Analysis of Opsins in Vertebrates

The amino acid sequences of the Japanese flounder and several other vertebrates (*Homo sapiens*, *Mus musculus*, *Gallus gallus*, *Xenopus tropicalis*, *Danio rerio*, *Hippoglossus hippoglossus* and *Cynoglossus semilaevis*) were analyzed for phylogenetic relationships. Evolutionary analyses were performed in MEGA 7.0 [35], where the evolutionary distances in the phylogenetic tree were calculated by a JTT matrix-based method [36] and the evolutionary history was inferred by the neighbor-joining method [37]. The reliability of the phylogenetic tree was assessed by a bootstrap test with 1000 replicates [38].

### 2.4. Quantitative Real-Time PCR 

Total RNAs were isolated from the larvae (including NC, TH, TU and rescue groups) and the adult tissues by using TRIzol Reagent (Invitrogen, Life Technology, Carlsbad, CA, USA). The processes were based on the manufacturer’s instructions. RNA integrity was detected by agarose gel electrophoresis, while the RNA concentration was detected using the spectrophotometer NANODROP 2000C (Thermo, Waltham, MA, USA), with a reading of 2.0 > A260/280 > 1.8. According to the manufacturer’s instructions, the total RNA (1 μg) of each sample was reverse-transcribed using an RT-PCR kit (Promega, Madison, WI, USA). 

We performed quantitative real-time PCR (qRT-PCR) analysis using the CFX96 Touch^TM^ Real-Time PCR Detection System (Bio-Rad, Hercules, CA, USA). The CDS for the opsin genes and TR (*trαa*, *trαb* and *trβ*) of the flounders were obtained from the NCBI website and were as follows: GI |109635199| (*rh*1); GI |109629258| (*lws*); GI |109638529| (*rh*2*b-*1); GI |109629165| (*sws*2*aβ*); GI |109641658| (*sws*1); GI |109641414| (*trαa*); GI |109643480| (*trαb*); GI |109631803| (*trβ*). Primer Premier 5.0 software was used to design primers (Table 1). The PCR reaction volume and conditions were set according to the instructions of the 2 × iQ™ SYBR green SuperMix (Bio-Rad, Hercules, CA, USA). Each experimental group underwent one set of repetition. The *β-actin* was used as the internal reference gene, and the relative expression levels of the five opsin genes were measured using the 2^−ΔΔCT^ method.

### 2.5. Enzyme-Linked Immunosorbent Assay (ELISA)

The TH levels in the flounders were determined by the enzyme-linked immunosorbent assay (ELISA) method. Frozen larvae samples from different metamorphosis periods (including NC, TH, and TU groups) were homogenized in pre-cooled phosphate buffer (PBS), and centrifuged at 5000× *g* for 10 min at 4 °C, and the supernatant was collected for further detection. T3 levels were determined using the Fish Triiodothyronine (T3) ELISA kit (HALING, Shanghai, China), referring to the manufacturer’s instructions for operation.

### 2.6. Promoter Region Prediction

Based on the flounder opsin genome sequences in the NCBI database, 2500 bp sequences of ATG upstream were selected for online promoter prediction analysis. The following two online promoter prediction websites were used to predict the promoter components: the Promoter 2.0 Prediction Server (https://services.healthtech.dtu.dk/service.php?Promoter-2.0), URL (accessed on 26 October 2022); and Softberry website (http://www.softberry.com/), URL (accessed on 26 October 2022). The Alibaba online database was used to analyze the potential transcription factor binding sites (http://gene-regulation.com/pub/programs/alibaba2/index.html), URL (accessed on 26 October 2022).

### 2.7. Double Luciferase Reporting Assay

The promoter sequences of the five opsin genes of the flounders were obtained from the NCBI database and used to design specific amplification primers (Table 2). The five promoter sequences were amplified by PCR and cloned into pGL3-basic vectors. The five resulting plasmids were then named pro-*rh*1, pro-*lws*, pro-*sws*2*aβ*, pro-*rh*2*b-*1 and pro-*sws*1, respectively. In addition, expression plasmids (p3×Flag-Trαa, p3×Flag-Trαb and p3×Flag-Trβ) were previously constructed and preserved in the laboratory. 293T cells were cultured in 10% FBS medium at 37 °C with 5% CO_2_. The cells were inoculated in 48-well cell culture plates the day before transfection, and transfection experiments were performed when the cell density reached about 70%. The constructed pGL3-pro-opsin recombinant plasmids, p3×Flag-TR expression plasmids, and internal reference plasmid pRL-TK were co-transfected into the 293T cells, and a group was set up to add T3 at a final concentration of 75 nM. After transfection was completed, the culture was continued for 24 h. The luciferase activity was detected according to the Dual Luciferase Reporter Assay System specification (Promega, Madison, WI, USA), and the ratio of Firefly Luciferase to Renilla Luciferase (*n* = 3) was recorded. 

### 2.8. Statistical Analysis

All data were expressed as means ± SE and analyzed by one-way ANOVA, and Dunnett’s post-hoc test for statistical differences, using SPSS 16.0 software. SigmaPlot 12.5 was used for graphs. *p <* 0.05 indicates a significant difference.

## 3. Results

### 3.1. Phylogenetic Analyses

Analysis for the phylogenetic tree (Figure 1) involved 43 amino acid sequences and the optimal tree with a total branch length of 5.60449067 is shown. The percentage of replicate trees in which the associated taxa clustered together in the bootstrap test (1000 replicates) is shown next to the branches [38]. The results show that flounders (*Paralichthys olivaceus*) and vertebrates cluster into a single branch and that flounder opsins are closely related to other *Osteichthyes*. Rh2b-1, Sws2aβ, Sws1 and Lws of flounders were most closely related to those of *Hippoglossus hippoglossus* (96%, 98%, 100% and 75%, respectively). In contrast, Rh1 in flounders is more distantly related to that of other species. 

### 3.2. Distribution of Five Opsin Genes in Adult Tissue

We used qRT-PCR to analyze the expression of the five opsin genes in adult tissues of the flounders (Figure 2). The genes were expressed in detectable but different quantities in the adult tissue. The levels of the five opsin genes (*rh*1, *lws*, *sws*2*aβ*, *rh*2*b-*1 and *sws*1) were all the highest in the eye tissue, being significantly higher than in other tissues (*p <* 0.05). In addition, there were no significant differences between the expressions of the five opsin genes in other tissues. 

### 3.3. Expression of Five Opsin Genes during Metamorphosis

As shown in the model diagram (Figure 3A), the flounder larvae gradually transitioned from the upper water to the lower water areas as the metamorphosis process occurred. The results of qRT-PCR showed multiple patterns of opsin genes expression during metamorphosis in the flounders. Among them, *rh*2*b-*1 was highly expressed from 17 dph to 20 dph, and then significantly decreased afterwards (*p <* 0.05) (Figure 3D). While the expression levels of the other four opsin genes did not change significantly between 17 dph and 20 dph, the levels of these four opsin genes started to increase rapidly after 20 dph. The expressions of the *rh*1 and *lws* genes reached their highest levels at 28 dph and then gradually decreased until 41 dph (Figure 3B,C). However, the expression of *sws*2*aβ* increased rapidly from 20 dph to 28 dph and remained high during the late metamorphosis stage (Figure 3E). The expression level of the *sws*1 gene peaked at 24 dph, decreased significantly afterwards (*p <* 0.05) and remained low during the middle and late stages of metamorphosis (Figure 3F). These results indicate that the expression of the five opsin genes varied with the metamorphosis process in flounders. 

### 3.4. Effect of Exogenous TH on Expression Levels of Opsins during Metamorphosis

As shown in Figure 4A, the flounder larvae’s metamorphosis proceeded normally in the normal control group (NC), was promoted in the TH-treated group (TH) and was inhibited in the TU-treated group (TU). In our study, it was found that exogenous T3 immediately increased T3 levels in eye tissue, peaking at 24 dph, while TU continuously inhibited T3 levels in the eyes during flounder metamorphosis (Figure 4B). As shown in Figure 4C, the expression levels of *trαa* in the eyes showed no significant differences among the four time points. On the other hand, the expression levels of *trαb* were significantly lower at 36 dph in the adult stage than that at 20 dph and 28 dph, while the levels of *trβ* were significantly higher at 28 dph, 36 dph in the adult stage than that at 20 dph (*p <* 0.05). Subsequently, we examined the relative expression levels of the opsin genes in the eyes of the NC, TH and TU groups during flounder metamorphosis. The following results were compared with the NC group. As shown in Figure 4D, the expression levels of *rh*1 in the eyes of TH-treated flounders were significantly up-regulated at 20 dph, 24 dph, 32 dph and 36 dph, but significantly down-regulated in the TU-treated flounders at 24 dph and 28 dph (*p <* 0.05). In Figure 4E, the *lws* levels in flounder eyes were significantly higher at 20 dph and 24 dph in the TH group and lower from 24 dph to 41 dph in the TU group (*p <* 0.05). In Figure 4F, the *rh*2*b-*1 levels in the flounder eyes significantly decreased from 17 dph to 28 dph in the TH group, while they were significantly up-regulated at 28 dph to 41 dph in the TU group (*p <* 0.05). As shown in Figure 4G, the expression levels of *sws*2*aβ* in the flounder eyes were significantly up-regulated at 17 dph, 20 dph and 24 dph in the TH group, but significantly down-regulated from 24 dph to 41 dph in the TU group (*p <* 0.05). In Figure 4H, the *sws*1 levels in flounder eyes were significantly down-regulated at 24 dph and 28 dph in the TH and TU groups, but up-regulated at 32 dph, 36 dph and 41 dph in the TU group (*p <* 0.05).

### 3.5. Expression Level of Opsins in the Rescue Larvae Inhibited by TU

To further investigate whether metamorphosis of the TU-treated flounders could be rescued, two subgroups of TU-treated larvae at 36 dph were moved into natural seawater (TU + NC group) and TH seawater (0.1 mg/L, TU + TH group), respectively. After 5 days of rearing, both of these groups of larvae successfully completed metamorphosis, while the larvae that continued to be treated with TU until 41 dph remained inhibited (Figure 5A). Meanwhile, expressions of the five opsin genes in different rescue groups were detected using qRT-PCR. As shown in (Figure 5B,D), the expression levels of *sws*2*aβ* and *lws* in the flounders were significantly higher in the rescue group compared to the TU group (*p <* 0.05), and were not significantly different from those of the NC and TH of 41 dph. The expressions of *sws*1 (Figure 5C) and *rh*2*b-*1 (Figure 5E) in the flounders in the rescue groups were significantly decreased (*p <* 0.05) compared with the TU group, and had no significant difference compared with the 41 dph NC and TH groups. The expression level of *rh*1 (Figure 5F) in the flounders had no significant difference between any of the groups. 

### 3.6. Targeted Regulation of Opsin Genes by T3 through TR

We investigated the targeting and regulatory relationship of T3 with the opsin genes through TRs by a dual luciferase reporter assay, for which five recombinant plasmids of the respective opsin promoter regions were co-transfected with p3×Flag-TR (p3×Flag-Trαa, p3×Flag-Trαb and p3×Flag-Trβ) recombinant plasmids. After normalization with data from each promoter activity group, the assay results show that the luciferase activity was significantly higher (*p* < 0.05) in the pro-*rh*1 + TRs group (*rh*1 promoter region plasmid co-transfected with p3×Flag-TRs), the pro-*lws* + Trαb/Trβ group and the groups with either of these plasmids and T3 simultaneously added (Figure 6A,B). When comparing the groups, the promoter activities of the pro-*sws*2*aβ* + Trαa + T3 group, the pro-*sws*2*aβ* + Trαb/Trβ group and the pro-*sws*2*aβ* + Trαb/Trβ + T3 group were significantly increased (*p* < 0.05) (Figure 6C). The promoter activities of the pro-*rh*2*b-*1 and pro-*sws*1 groups were not significantly different in any of the groups (Figure 6D,E).

## 4. Discussion

Fish have evolved over time to adapt their visual systems to changing light environments, and have increased their opsin diversity through the replication and divergence of opsins [39]. The Japanese flounder was found to have ten opsin genes, including three blue opsin genes and four green opsin genes (*rh*1, *lws*, *sws*1, *sws*2*aα*, *sws*2*aβ*, *sws*2*b*, *rh*2*a-*1, *rh*2*a-*2, *rh*2*b-*1 and *rh*2*b-*2). In addition, the flounders adapted to change in their light environment during development by adjusting their *sws*2*a* and *rh*2 genes, with *sws*2*aβ* showing the most significant elevation in expression during metamorphosis and *rh*2*b-*1 maintaining its original green-sensitive function during metamorphosis with a gradual decrease in expression level, consistent with the transition from the upper water to the lower water areas during metamorphosis [9]. Similar studies have found that turbot (*Scophthalmus maximus*) have two blue opsin genes and five green opsin genes, and barfin flounders (*Verasper moseri*) have two blue opsin genes and three green opsin genes [10,11]. They undergo dramatic changes in their light environment during growth and development in a similar manner to Japanese flounders, and the expression levels of their opsins also significantly change.

In our current study, we found that all five opsin genes (*rh*1, *lws*, *sws*1, *sws*2*aβ* and *rh*2*b-*1*)* of flounders are up to hundreds of times more expressed in eye tissues, even hundreds of times more than in other tissues. Similarly, the expression of the opsin genes of mudskipper and crimson snapper fish have been found to be significantly higher in the eye tissues than in other tissues [40,41]. Related studies have demonstrated that opsins are mainly distributed in visual tissues such as the retina, which act as photoreceptors in animals and participate in their adaptation to external light environments [42]. Subsequently, we found that the expression of the *sws*2*aβ* gene peaked at mid-metamorphosis and remained high during metamorphosis, while the expression of the other four opsin genes showed a trend of increasing and then decreasing. Related studies have shown that the diversity of the types and expression levels of fish opsin genes are closely related to the light environment of the water body in which they live [43,44,45]. It has been found that red grouper and medaka can change the expression levels of their opsins to adapt to living in different light environments [46,47]. In our study, the flounder larvae were active in the upper water layer at the beginning of metamorphosis and gradually transitioned to deeper blue-dominated waters during metamorphosis, which corresponds to the result of the high expression levels of their *sws*2*aβ* genes at the middle and end of metamorphosis. Therefore, our results further suggest that the differential expression of opsin genes is closely related to the metamorphosis process in flounders. 

In *Paralichthys olivaceus*, metamorphosis is accelerated by TH [18], but inhibited by TU [33]. We found that the T3 levels increased at the beginning of metamorphosis for TH-treated larvae, but were significantly decreased throughout the metamorphosis period for TU-treated larvae (*p* < 0.05). Meanwhile, we found that the levels of the *lws* and *sws*2*aβ* genes were rapidly increased by TH treatment and decreased after TU treatment. In addition, TU-suppressed flounder larvae could be rescued by natural seawater or exogenous TH and proceed to metamorphose normally. Moreover, the expressions of the *sws*2*aβ* and *lws* genes were significantly increased in the rescue group compared with the TU group (*p* < 0.05). This further suggests that TH influences the metamorphosis process of flounder larvae, which in turn affects the expression levels of opsins during flounder metamorphosis. 

TH and its receptor (TR) play a role in regulating photoreceptor differentiation and opsin expression not only in the flounder, but also in the retinas of rodents and other fishes [48]. Studies in rats first demonstrated that retinal thickness and cell number are reduced due to congenital hypothyroidism [49]. TH is important for the differentiation of L-cone as opposed to UV cone fate in zebrafish [50]. We found that the levels of three isoforms of TR (*trαa*, *trαb* and *trβ*) were differently expressed at various periods of flounder growth, with *trβ* being highly expressed in the middle and late stages of metamorphosis. Similarly, *trαa*, *trαb* and *trβ* are all expressed in the retina of zebrafish, with *trβ* demonstrating the highest expression [51]. Thyroid hormone receptors similarly affect the fate of photoreceptors, and one study has found that Trβ affects the number of blue cones in zebrafish and that a decrease in *trβ*2 in zebrafish leads to an increase in the number of cones that express photoreceptors, while mutations in *trβ* lead to the loss of red cones in juvenile fish and their conversion to UV cones and horizontal cells [52,53,54,55]. In mice, the specific knockout of *Trβ* has resulted in a selective loss of M-cones and an increase in S-cones. The expanded distribution of the S-type cones in the retina and the increase in S-cones in adult mice suggest that TRβ2 can inhibit S-opsin until an appropriate stage of development [32,56], which has also been observed in other rodents [51]. In zebrafish, mice and humans, T3 signaling drives specific opsin expression through the thyroid hormone receptor β2, which is expressed only in the cones of the retina [57,58]. By verifying the targeting regulatory relationship of T3 through TRs in opsin genes, we found that T3 and TRs positively regulate *rh*1, *lws* and *sws*2*aβ* genes. It is hypothesized that T3 may directly or indirectly regulate the expression of the *sws*2*aβ* promoter through Trαa, which may be closely related to the blue-dominated benthic life of flounders. This hypothesis needs further experimental verification in future studies.

## 5. Conclusions

Our experimental results demonstrate that the differential expression of opsins in flounders is closely related to the light environment during metamorphosis, and verify that TH can positively regulate the expression of *rh*1, *lws* and *sws*2*aβ* in flounders, and may even directly or indirectly regulate the expression of *sws*2*aβ* through Trαa. It is further demonstrated that TH plays an important role in the development of the visual photosensitive system during the metamorphosis of flounders.

## Figures and Tables

**Figure 1 biology-12-00397-f001:**
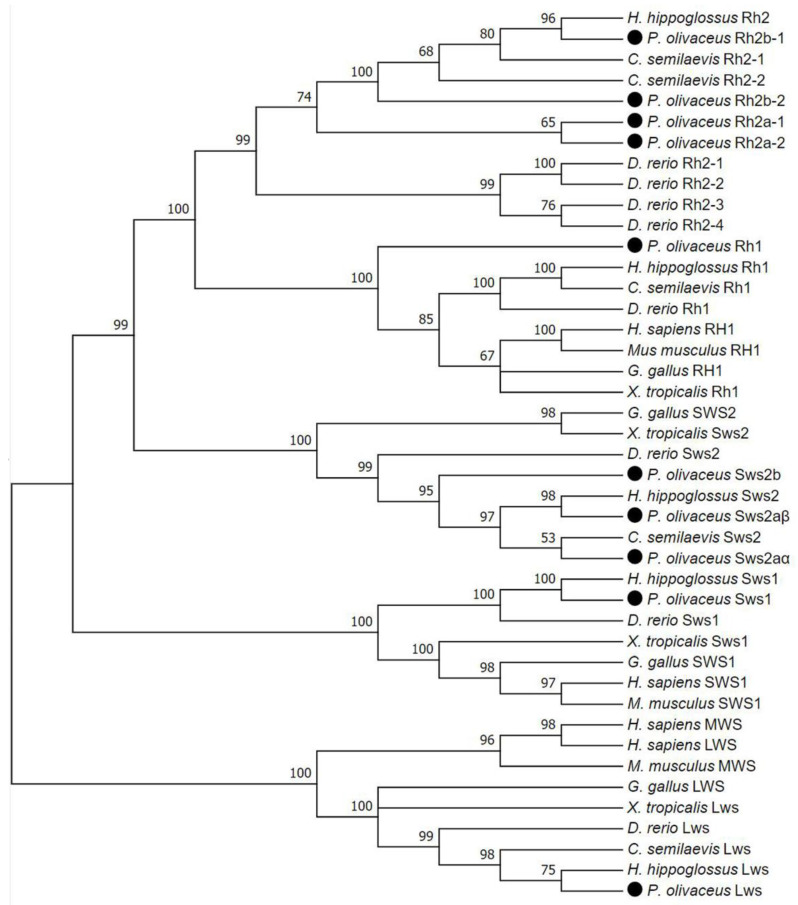
Evolutionary relationships of taxa. The accession numbers of the amino acid sequences of the five genes for analysis are as follows: *Homo sapiens* (RH1: NP_000530.1; MWS: NP_000504.1; LWS: NP_064445.2; SWS1: NP_001372054.1); *Mus musculus* (RH1: NP_663358.1; MWS: NP_032132.1; SWS1: NP_031564.1); *Gallus gallus* (RH1: NP_001384426.1; LWS: NP_990771.2; SWS2: NP_990848.1; SWS1: NP_990769.1); *Xenopus tropicalis* (Rh1: NP_001090803.1; Lws: NP_001096331.1; Sws2: XP_002937272.2; Sws1: NP_001119548.1); *Danio rerio* (Rh1: NP_571159.1; Rh2-1: NP_571328.2; Rh2-2: NP_878311.1; Rh2-3: NP_878312.1; Rh2-4: NP_571329.1; Lws: NP_001300644.1; Sws2: NP_571267; Sws1: NP_571394.1); *Hippoglossus hippoglossus* (Rh1: XP_034445498.1; Rh2: AAM17916.1; Lws: XP_034441780.1; Sws2: AAM17920.1; Sws1: AAM17917.1); *Cynoglossus semilaevis* (Rh1: XP_024915364.1; Rh2-2: XP_008317993.1; Rh2-1: XP_008317992.1; Lws: XP_008318886.1; Sws2: XP_008318880.1); *Paralichthys olivaceus* (Rh1: XP_019951764.1; Rh2b-1: BAW35578.1; Rh2a-1: BAW79257.1; Rh2b-2: BAW35580.1; Rh2a-2: BAW79258.1; Lws: XP_019942446.1; Sws2aβ: BAW35586.1; Sws2aα: XP_019942172.1; Sws2b: BAW35589.1; Sws1: XP_019961738.1).

**Figure 2 biology-12-00397-f002:**
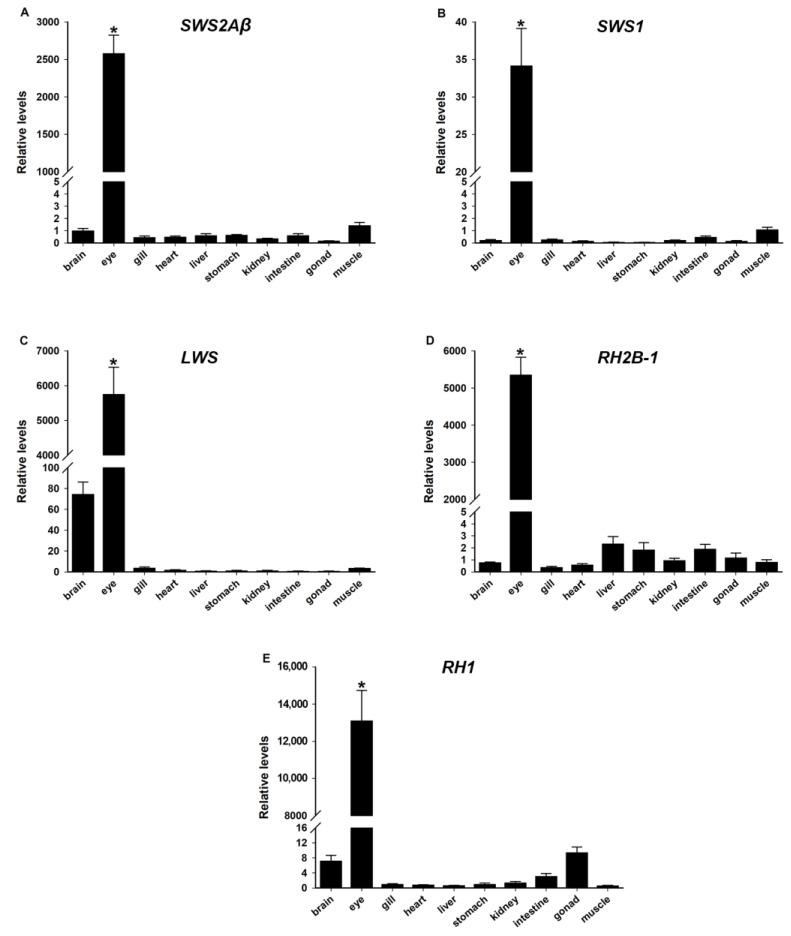
Distribution of five opsin genes in adult tissue of flounders. (**A**) *sws*2*aβ*. (**B**) *sws*1. (**C**) *lws*. (**D**) *rh*2*b-*1. (**E**) *rh*1. Error bars indicate the mean ± standard error (SE) of each value expression (*n* = 3). The *sws*2*aβ* level in the brain was used as a reference. The asterisk indicates a significant difference from the brain group (*p* < 0.05).

**Figure 3 biology-12-00397-f003:**
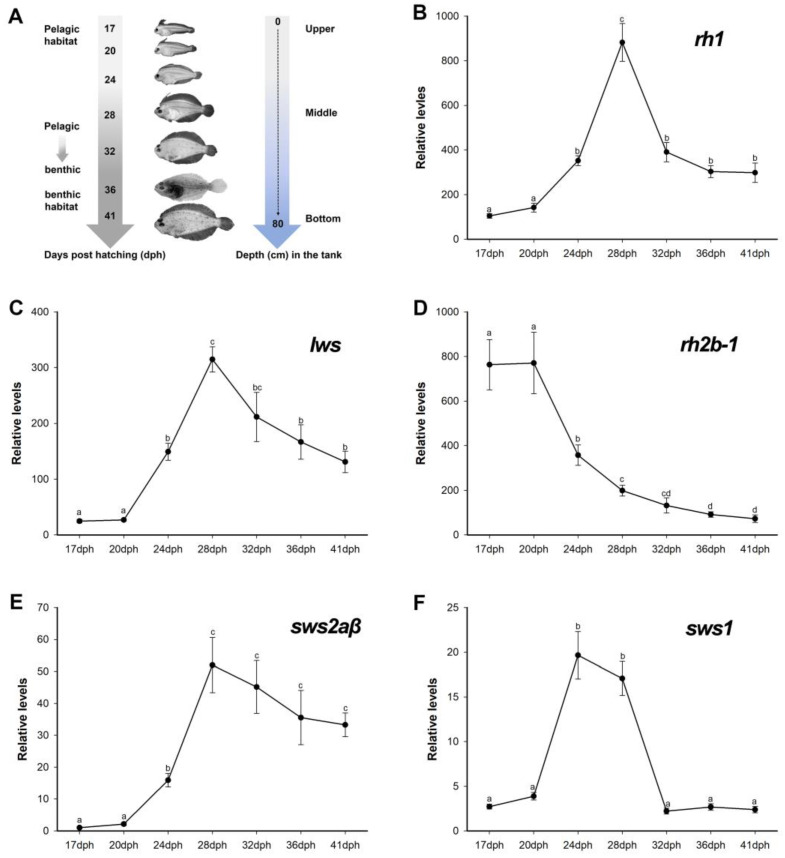
Expression patterns of five opsin genes during metamorphosis in flounders. (**A**) Diagram of a flounder’s migration pattern through its environment during metamorphosis. (**B**) *rh*1. (**C**) *lws*. (**D**) *rh*2*b-*1. (**E**) *sws*2*aβ*. (**F**) *sws*1. The mean ± standard error (SE) of each value (*n* = 3) is shown by an error bar. The *sws*2*aβ* level at 17 dph was used as a reference. Different letters indicate groups with significant differences (*p* < 0.05) between them.

**Figure 4 biology-12-00397-f004:**
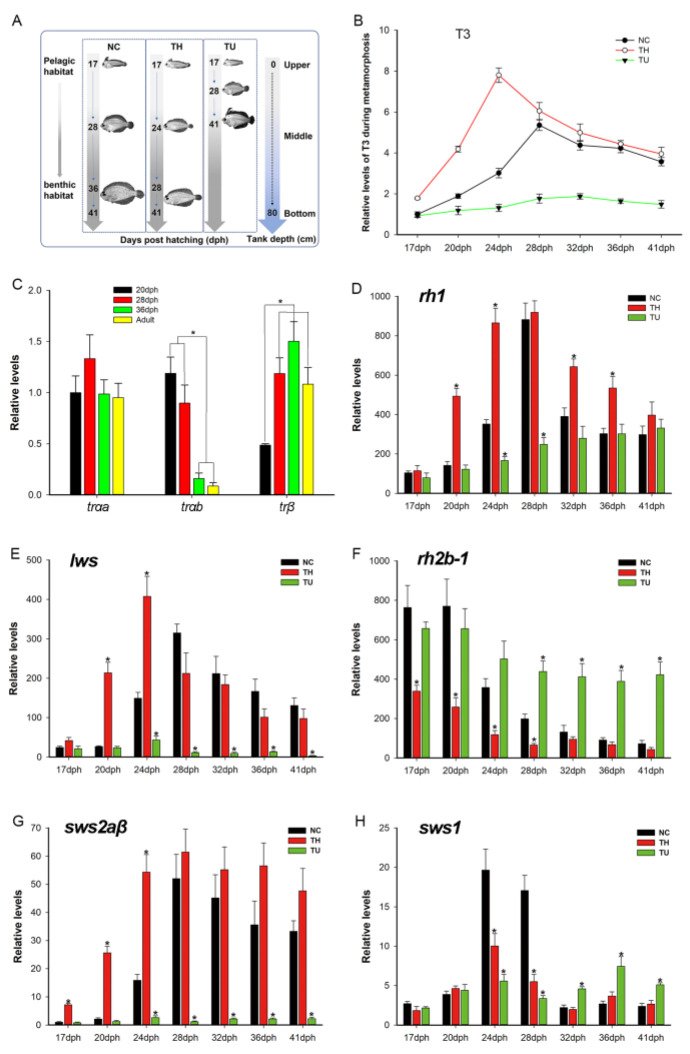
T3 levels and expression levels of five opsin genes in TH- and TU-treated flounders. (**A**) Metamorphosis pattern of flounders in TH and TU treatment groups. (**B**) T3 levels during metamorphosis of flounders under exogenous TH and TU treatment. (**C**) Expression of TRs in the eyes at different periods. (**D**) *rh*1. (**E**) *lws*. (**F**) *rh*2*b-*1. (**G**) *sws*2*aβ*. (**H**) *sws*1. The mean ± standard error (SE) of each value (*n* = 3) is shown by an error bar. The *sws*2*aβ* level at 17 dph for the NC group was used as a reference. The asterisks indicate significant differences (*p* < 0.05) from the contemporaneous NC group.

**Figure 5 biology-12-00397-f005:**
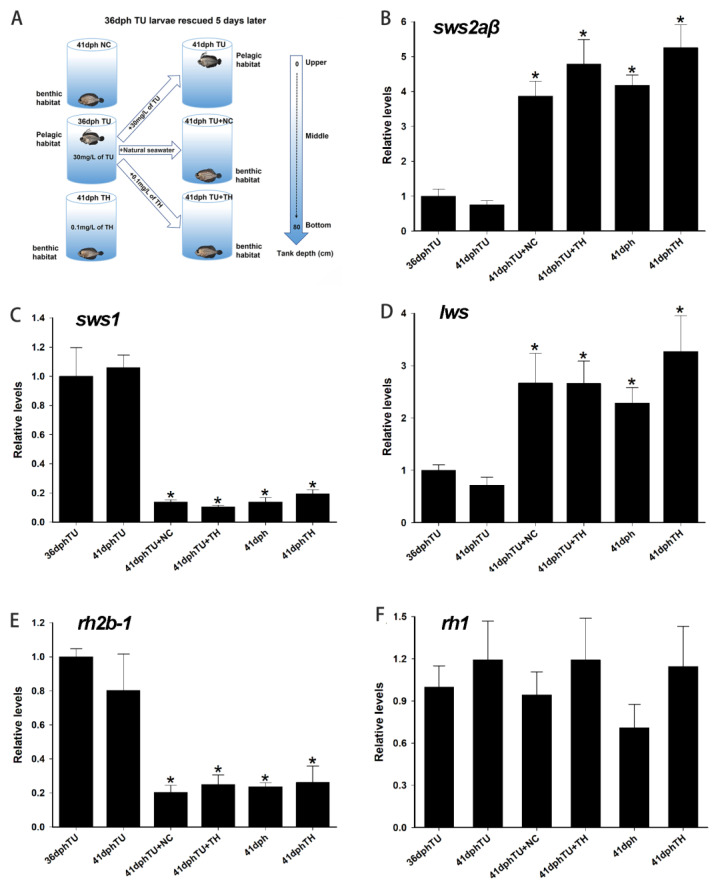
Relative expression levels of opsin genes in the larvae rescued from inhibition by TU. (**A**) Metamorphosis pattern of flounder larvae in the rescue group. (**B**) *sws*2*aβ*. (**C**) *sws*1. (**D**) *lws*. (**E**) *rh*2*b-*1. (**F**) *rh*1. The 41 dph TU + NC and 41 dph TU + TH groups labels indicate 36 dph TU-treated larvae that were transferred to natural seawater and seawater supplemented with 0.1 mg/L T3, respectively, and reared there until 41 dph. The 36 dph TU level was used as a reference. The mean ± standard error (SE) of each value (*n* = 3) is shown by an error bar. The asterisks indicate significant differences (*p* < 0.05) from the TU-treated group (TU).

**Figure 6 biology-12-00397-f006:**
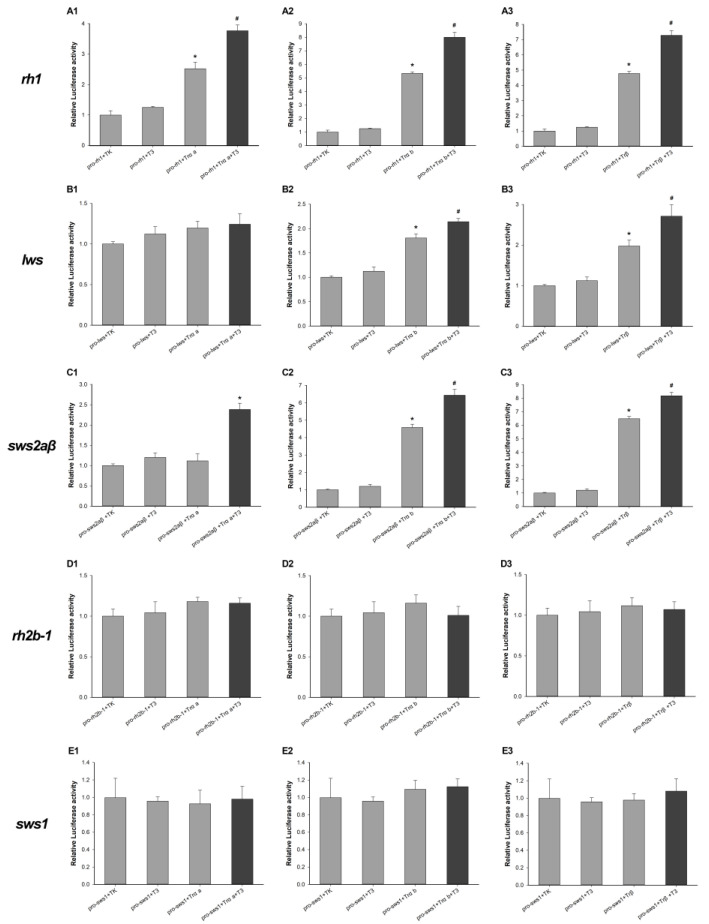
Targeted regulation of opsin genes by T3 through TR. (**A1**–**A3**) *rh*1. (**B1**–**B3**) *lws*. (**C1**–**C3**) *sws*2*aβ*. (**D1**–**D3**) *rh*2*b-*1. (**E1**–**E3**) *sws*1. The mean ± standard error (SE) of each value (*n* = 3) is shown by error bars. Data from each promoter activity group were used for normalization. The asterisks indicate significant differences (*p* < 0.05) from the values in the promoter activity group and the other different symbols indicate groups with significant differences (*p* < 0.05) between them.

**Table 1 biology-12-00397-t001:** Primers used in the experiment.

Primer Name ^1^	Primer Sequence (5′–3′)	Application
*rh*1-F	AGGGCTCTGAGTTCGGAC	qRT-PCR
*rh*1-R	TCGGTCTTGGTGCTGGAT	qRT-PCR
*lws*-F	GCAGAAGGAATCAGAGTCAAC	qRT-PCR
*lws*-R	ATGTGCGGAACTGTCGGT	qRT-PCR
*sws*2*aβ*-F	TGTGGGCACTAGCATCAAC	qRT-PCR
*sws*2*aβ*-R	CAGCAGCCAACAAAGGAG	qRT-PCR
*rh*2*b-*1-F	GGAAGCCTTGTGCTGACA	qRT-PCR
*rh*2*b-*1-R	ACGAGGAAGCCAATGACC	qRT-PCR
*sws*1-F	CTCCTGTGGTCCTGATTG	qRT-PCR
*sws*1-R	ATGATGATGCTGAGTGGG	qRT-PCR
*trαa*-F	AAAACCTCCACCCATCTTACTCC	qRT-PCR
*trαa*-R	GTCCGCACCATCTCCTCCC	qRT-PCR
*trαb*-F	GCATTACTTGCGAGGGC	qRT-PCR
*trαb*-R	AATAGTGAAAACCCTCCAGA	qRT-PCR
*trβ*-F	GCCTTGAACCCCACCAGTATG	qRT-PCR
*trβ*-R	AGGGTTTCTTCAGGCGGACA	qRT-PCR
*β-actin*-F	GGAAATCGTGCGTGACATTAAG	qRT-PCR
*β-actin*-R	CCTCTGGACAACGGAACCTCT	qRT-PCR

^1^ F: forward primers. R: reverse primers.

**Table 2 biology-12-00397-t002:** Primers used in the experiment.

Primer Name ^1^	Primer Sequence (5′–3′) ^2^	Application
pro-*rh*1-F	atctgcgatctaagtaagcttTTTGAACACTTTCACTCTTAGAAAAGTCT	Promoter amplification
pro-*rh*1-R	cagtaccggaatgccaagcttGGCTGCTGACGGTGATGGG	Promoter amplification
pro-*lws*-F	atctgcgatctaagtaagcttAATGAGTATATGTTTTGCAAGCACTTC	Promoter amplification
pro-*lws*-R	cagtaccggaatgccaagcttTTTGTTCTTAGCAGGAGGGCC	Promoter amplification
pro-*sws*2*aβ*-F	atctgcgatctaagtaagcttCCTTAAACAAATTACAAACCACGACG	Promoter amplification
pro-*sws*2*aβ*-R	cagtaccggaatgccaagcttTTTTTCCCCCACGGGCAA	Promoter amplification
pro-*rh*2*b-*1-F	atctgcgatctaagtaagcttAGAGAGTCCGGGAACAATAGCA	Promoter amplification
pro-*rh*2*b-*1-R	cagtaccggaatgccaagcttCTTCGATTGTCTGTTGTTTCTGCT	Promoter amplification
pro-*sws*1-F	atctgcgatctaagtaagcttGAGGAGTGGAAGAGTGAAGGAGTAG	Promoter amplification
pro-*sws*1-R	cagtaccggaatgccaagcttGAACCTGAGCTTTCTAACACTAGGACG	Promoter amplification

^1^ F: forward primers. R: reverse primers. ^2^ Primer Premier 5.0 was used to design the primers.

## Data Availability

All data generated in this study are included in the article. Additional information can be requested from the corresponding authors if necessary.

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
