# Peer review of "Thyroid Hormone Signaling Is Required for Dynamic Variation in Opsins in the Retina during Metamorphosis of the Japanese Flounder (Paralichthys olivaceus)"

_biology, 2023, doi:10.3390/biology12030397_

Round 1
Reviewer 1 Report (Previous Reviewer 3)
During larval development, flounders transform from fish freely swimming in the water column to bottom dwellers. Shi et al show that this process is mediated by thyroid hormone T3. Adding thyroid hormone to the water accelerates the transition, while adding phenylthiourea (PTU), an inhibitor of thyroid hormone synthesis, prevents the process. The thyroid hormone system also mediates changes in the expression of five of the ten opsin genes found in Japanese flounder. In particular the LWS (red) opsin gene and the SWS2Abeta gene, one of three blue opsin genes, increase in activity, while the RH2B-1 gene decreases in activity. These changes are thought to be adaptations to the deep-water spectral environment. The involvement of three of the thyroid hormone receptors is identified.
The reviewer wasn’t convinced by Fig. 6. This involved ‘periocular’ injections of PTU to inhibit production of thyroid hormone. The hormone isn’t synthesized in the eye, but the thyroid follicles, whose proximity to the injection site was not clear. How long this inhibitor would persist in the fish circulation was also not clear, so the reviewer didn’t know whether thyroid hormone was reduced by this method. The Figure could be removed without compromising the paper.
English usage was poor in many areas, and the reference section was unacceptably poorly done.
Minor comments:
49 ‘and different types of photoreceptor (PR) subtypes within the retina’; and different photoreceptor (PR) subtypes
50 ‘Most fish contain two PR types: rod photo-receptor and cone photoreceptor.’; rod photoreceptors and cone photoreceptors
51 ’The rod secretes Rhodopsin (RH1),’ ‘the cone secretes red-sensitive opsin’; poor word usage, secretion is for chemicals synthesized and sent outside the cell, as in ‘the pancreas secretes insulin’. Change to ‘the rod synthesizes rhodopsin…’
56 ’Visual systems of fish are usually complex’; unusually complex
65 ’which characterized by body shape from symmetrical to asymmetrical, migration of eye from right to left, and transition of habitat from pelagic to benthic which lead to changes from light to dim in the photic environment [17].’ Lots of grammar problems. Change to ‘which is characterized by change in body shape from symmetrical to asymmetrical, migration of an eye from right to left, and transition of habitat from pelagic to benthic which leads to changes from bright to dim in the photic environment [17].’
74 ’the main form of action is triiodothyronine (T3)’ ‘the active form is triiodothyronine (T3)’
92 ’ of Chinese Academy of Fishery Sciences’ ; of the Chinese Academy of Fishery Sciences
93 ’The flounder were intensively culture in seawater at 19-21℃’; …cultured in seawater…
362’ the TU group cultured in seawater’; …was cultured…
69 ’The TU group was continued to be cultured’; The TU group continued to be cultured
140 ’ Each experimental group did one set of repetitions.’ The reader doesn’t understand the sentence.
228 Fig. 2. The legend should say ‘adult tissue’
245 Fig. 3. The units are given as relative, are these also relative to brain SWS2Abeta? While I am not an expert on qPCR, I have heard the primers may differ in efficiency, and therefore, would it be safer to normalize each plot to a time point in the plot? In other words, to what extent does a qPCR ratio predict relative opsin production? What are the meanings of ‘a’, ‘b’,’c’,’d’, ‘bc’, and ‘cd’ in the plots. The figure legend should define these, as well as the normalization. It would seem that there could be several hypotheses about the gene expression profiles: growth of outer segments, increase in the number of cells, changes in the rate of disk shedding, ie turnover. What do these plots say?
256 ’ As showed in (Figure 4C), the expression levels of TRαA in the eyes were no significant difference among the four time points’. As shown in Fig. 4C, the eye expression levels of TrαA did not differ significantly at the four time points.
259 ‘while the levels of TRβ was significant higher’ were significantly higher
264 ‘while it was significantly down-regulated in the’ while they were significantly down-regulated
268 ‘while it was significantly up-regulated at 28 dph to 41 dph in the TU group’ while they were
277 ‘T3 levels and expression levels of five gene opsins’ five opsin genes
279 ‘Expression of TRs in the eyes of different periods.’ Expression of TRs during eye development
296 ‘Expression level of RH1 (Figure 5F) in the flounder was no significant’ not significant
299 Fig. 5A, The figure would be less ambiguous if the ‘41 dph TU+NC’ treatment was named ’41 dph NC’ and the ‘41dph TU+TH’ treatment was named ’41 dph TH’. The treatment did not include both TU and TH. To be more explicit one could say ‘TU⇨TH’ (the character is an open arrow).
316 Fig. 6. The reviewer was not convinced that Fig. 6 was needed. A skeptical reader might conclude that the ‘periocular injection’ method didn’t deliver sufficient TH or TU to the tissue.
384 ‘In contrast, there was no significant decrease in the expression of all opsin genes after TU injection.’ TU inhibits TH synthesis, but this synthesis does not occur in periocular tissue, but in the thyroid follicles (Elsalini et al, 2003). The reviewer doesn’t know if the PTU injection were near the thyroid follicles, and what the pharmacokinetics might be for PTU.
439 References are full of errors, particularly in correctly citing author names.
Some examples:
448 ‘Whitmore; David; Josephine K., et al. An extended family of novel vertebrate photopigments is widely expressed and displays a diversity of function. 25(11): 1666-1679’
This would appear to show incorrect authors and not list any journal. This is what the reviewer finds.
Davies WI, Tamai TK, Zheng L, Fu JK, Rihel J, Foster RG, Whitmore D, Hankins MW. An extended family of novel vertebrate photopigments is widely expressed and displays a diversity of function. Genome research. 2015 Nov 1;25(11):1666-79.
458 ‘Chen; Songlin; Zhang, et al.’ Chen S, Zhang G et al
460 ‘M M.M. and A C.D.’ Mader MM, Cameron DA
462 ‘V H.J.; O D.; H N.T., et al.’ Helvik JV, Drivenes Ø et al
468 ‘W I.T.; Iñigo N.F.; Juan A., et al.’ Iwanicki TW, Novales Flamarique I et al
475 ‘Jeffrey T.; Sanjiv H.; Nathan B., et al.’ Trimarchi JM, Harpavat S et al
497 ‘L N.; B H.J.; B D., et al.’ Ng, L, Hurley, JB et al
505 ‘T J.D.; R T.W. and M T.J.’ Jones, DT, Taylor, WR and Thornton, JM
507 ‘N S. and M N.’ Saitou, N and Nei, M
511 ‘Cameron W. and Belinda C.’ Weadick, CJ and Chang, BS
514 ‘Qiulu L.; Gyamfua A.; Zizhao C., et al.’ Q Liang, G Afriyie et al
517 ‘A T.; S K.; C v.O., et al.’ Tezuka, A., Kasagi, S. et al
523 ‘C F.R. and M C.K.’ Fuller, RC and Claricoates, KM
531 ‘A S.; A L.; T F., et al.’ Szél, Á, Lukáts, Á et L
533 ‘Annastelle C.; Jeremy P.; Mikayla D.P., et al.’ Cohen A, Popowitz J et al
535 ‘I V.L.; Sook K.-H.J.; M S.L.’ Volkov, LI, Kim-Han, JS et al
537 ‘F N.R.; Annika B.; Tara S., et al.’ Nelson, RF, Balraj, A et al
540 ‘Ciana D.; Xiaodong J.; C S.S., et al.’ Deveau, C, Jiao, X et al
542 ‘A S.; P R.; K M., et al.’ Szel, A, Röhlich, P et al
545 ‘D M.R.; A F.R.; Carmina G., et al.’ Mackin, RD, Frey, RA
Author Response
Thank you for your patient review! We have carefully revised the English language and references of this manuscript. Please see the attachment, and the corresponding changes have been marked in the manuscript.
Point 1: The reviewer wasn’t convinced by Fig. 6. This involved ‘periocular’ injections of PTU to inhibit production of thyroid hormone. The hormone isn’t synthesized in the eye, but the thyroid follicles, whose proximity to the injection site was not clear. How long this inhibitor would persist in the fish circulation was also not clear, so the reviewer didn’t know whether thyroid hormone was reduced by this method. The Figure could be removed without compromising the paper.
Point 23: 316 Fig. 6. The reviewer was not convinced that Fig. 6 was needed. A skeptical reader might conclude that the ‘periocular injection’ method didn’t deliver sufficient TH or TU to the tissue.
Point 24: 384 ‘In contrast, there was no significant decrease in the expression of all opsin genes after TU injection.’ TU inhibits TH synthesis, but this synthesis does not occur in periocular tissue, but in the thyroid follicles (Elsalini et al, 2003). The reviewer doesn’t know if the PTU injection were near the thyroid follicles, and what the pharmacokinetics might be for PTU.
Response 1, 23 and 24: We have carefully considered your suggestion and decided to remove Figure 6 and the related content.
Point 2: 49 ‘and different types of photoreceptor (PR) subtypes within the retina’; and different photoreceptor (PR) subtypes
Response 2:This sentence is corrected to ‘and different photoreceptor (PR) subtypes within the retina.’
Point 3: 50 ‘Most fish contain two PR types: rod photo-receptor and cone photoreceptor.’; rod photoreceptors and cone photoreceptors
Response 3:This sentence is corrected to ‘Most fish contain two PR types: rod photoreceptors and cone photoreceptors.‘
Point 4: 51 ’The rod secretes Rhodopsin (RH1),’ ‘the cone secretes red-sensitive opsin’; poor word usage, secretion is for chemicals synthesized and sent outside the cell, as in ‘the pancreas secretes insulin’. Change to ‘the rod synthesizes rhodopsin…’
Response 4:This sentence is corrected to ‘The rod synthesizes Rhodopsin (RH1), which is responsible for vision in low light, and the cone synthesizes red-sensitive opsin (M/LWS).’
Point 5: 56 ’Visual systems of fish are usually complex’; unusually complex
Response 5:This sentence is corrected to ‘Visual systems of fish are unusually complex.‘
Point 6: 65 ’which characterized by body shape from symmetrical to asymmetrical, migration of eye from right to left, and transition of habitat from pelagic to benthic which lead to changes from light to dim in the photic environment [17].’ Lots of grammar problems. Change to ‘which is characterized by change in body shape from symmetrical to asymmetrical, migration of an eye from right to left, and transition of habitat from pelagic to benthic which leads to changes from bright to dim in the photic environment [17].’
Response 6:This sentence is corrected to ‘characterized by change of body shape from symmetrical to asymmetrical, migration of the right eye wards to the left, as well as transition from pelagic to benthic habitat, and thus from light to dim photic environments [17]..’
Point 7: 74 ’the main form of action is triiodothyronine (T3)’ ‘the active form is triiodothyronine (T3)’
Response 7:This sentence is corrected to ‘the active form is triiodothyronine (T3)’
Point 8: 92 ’ of Chinese Academy of Fishery Sciences’ ; of the Chinese Academy of Fishery Sciences
Response 8:This sentence is corrected to ‘of the Chinese Academy of Fishery Sciences’
Point 9: 93 ’The flounder were intensively culture in seawater at 19-21℃’; …cultured in seawater…
Response 9:This sentence is corrected to ‘The flounder were intensively cultured in seawater at 19-21℃.’
Point 10: 362 ’the TU group cultured in seawater’; …was cultured…
Response 10:This sentence is corrected to ‘the TU group was cultured in seawater.’
Point 11: 69 ’The TU group was continued to be cultured’; The TU group continued to be cultured
Response 11:This sentence is corrected to ‘a TU group continued to be cultured.’
Point 12: 140 ’ Each experimental group did one set of repetitions.’ The reader doesn’t understand the sentence.
Response 12:This sentence is corrected to ‘Each experimental group had one set of repetition.’
Point 13: 228 Fig. 2. The legend should say ‘adult tissue’
Response 13:This sentence is corrected to ‘Figure 2. Distribution of five opsin genes in adult tissue of flounder.’
Point 14: 245 Fig. 3. The units are given as relative, are these also relative to brain SWS2Abeta? While I am not an expert on qPCR, I have heard the primers may differ in efficiency, and therefore, would it be safer to normalize each plot to a time point in the plot? In other words, to what extent does a qPCR ratio predict relative opsin production? What are the meanings of ‘a’, ‘b’,’c’,’d’, ‘bc’, and ‘cd’ in the plots. The figure legend should define these, as well as the normalization. It would seem that there could be several hypotheses about the gene expression profiles: growth of outer segments, increase in the number of cells, changes in the rate of disk shedding, ie turnover. What do these plots say?
Response 14:In Figure 3, we normalized the data with SWS2Aβ gene at 17 dph so that we could not only know the expression changes of these genes during the metamorphosis of dental flounder, but also briefly understand the differences in expression levels between different genes. And these different letters are used to indicate statistical differences between data and are described in the legend ‘Different letters indicate groups with significant differences (P<0.05) between them.’ In addition, Figure 3 depicts the changes in the expression levels of the five opsin genes during metamorphosis in flounder, i.e., the temporal expression profiles of the five opsin genes during metamorphosis in flounder.
Point 15: 256 ’ As showed in (Figure 4C), the expression levels of TRαA in the eyes were no significant difference among the four time points’. As shown in Fig. 4C, the eye expression levels of TrαA did not differ significantly at the four time points.
Response 15:This sentence is corrected to ‘As shown in Figure 4C, the expression levels of TRαA in the eyes showed no significant differences among the four time points.’
Point 16: 259 ‘while the levels of TRβ was significant higher’ were significantly higher
Response 16:This sentence is corrected to ‘while the levels of TRβ were significantly higher.’
Point 17: 264 ‘while it was significantly down-regulated in the’ while they were significantly down-regulated
Response 17:This sentence is corrected to ‘but significantly down-regulated in the’
Point 18: 268 ‘while it was significantly up-regulated at 28 dph to 41 dph in the TU group’ while they were
Response 18:This sentence is corrected to ‘while they were significantly up-regulated at 28 dph to 41 dph in the TU group.’
Point 19: 277 ‘T3 levels and expression levels of five gene opsins’ five opsin genes
Response 19:This sentence is corrected to ‘T3 levels and expression levels of five opsin genes.’
Point 20: 279 ‘Expression of TRs in the eyes of different periods.’ Expression of TRs during eye development
Response 20:This sentence is corrected to ‘Expression of TRs in eyes of different metamorphic stages.’
Point 21: 296 ‘Expression level of RH1 (Figure 5F) in the flounder was no significant’ not significant
Response 21:This sentence is corrected to ‘Expression level of RH1 (Figure 5F) in the flounder had no significant difference between all groups.’
Point 22: 299 Fig. 5A, The figure would be less ambiguous if the ‘41 dph TU+NC’ treatment was named ’41 dph NC’ and the ‘41dph TU+TH’ treatment was named ’41 dph TH’. The treatment did not include both TU and TH. To be more explicit one could say ‘TU⇨TH’ (the character is an open arrow).
Response 22: In Figure 5 we include the results of not only the TU treatment group, TU+NC and TU+TH rescue groups, but also the NC and TH treatment groups. We use '+' to indicate that we cultured in TU-added seawater until 36 dph and then moved to normal (NC) seawater and TH-added (TH) seawater after 36 dph to continue the culture until 41 dph.

Reviewer 2 Report (Previous Reviewer 2)
I think the authors answered most of the questions.
Major points;
· The manuscript requires careful editing for English language usage. I strongly recommend that the authors use a professional language service.
· There are multiple mistakes in the reference section. The authors must correct them carefully. For example, the journal name of reference #5 is wrong (Line 445).
· Response 15: Fetal Bovine Serum (FBS) contains T3, meaning a significant amount of T3 is already included in the culture medium, even if you do not add exogenous T3. The author's response doesn't answer the question.
Minor points;
· Line 51, Line 52: In Japanese Flounder, do photoreceptors "secrete" opsin photopigments?
· Lines 78-81: This description is about the mouse retina. It's better to state it clearly.
· In Figure 1, what is the P. olivaceus "SWS2A(2)" ?
Author Response
Thank you for your patient review! We have carefully revised the English language and references of this manuscript. Please see the attachment, and the corresponding changes have been marked in the manuscript.
Point 1: · There are multiple mistakes in the reference section. The authors must correct them carefully. For example, the journal name of reference #5 is wrong (Line 445).
Response 1:This sentence is corrected to ‘Bowmaker J.K. and Hunt D.M. Evolution of vertebrate visual pigments. Current Biology, 2006. 16(13).’
Point 2: · Response 15: Fetal Bovine Serum (FBS) contains T3, meaning a significant amount of T3 is already included in the culture medium, even if you do not add exogenous T3. The author's response doesn't answer the question.
Response 2:The results of luciferase analysis showed that the addition of TRαB/TRβ group and T3+TRαB/TRβ group could up-regulate the expression of SWS2Aβ. For this result, we have two speculations: 1. endogenous T3 regulates the expression of SWS2Aβ through TRαB and TRβ, and 2. TRαB and TRβ act as transcription factors to directly regulate the expression of SWS2Aβ. Therefore, a direct relationship between T3 and TRαB/TRβ could not be demonstrated. However, the difference in luciferase activity of SWS2Aβ in the added TRαA group was not significant, and we concluded that TRαA was not acting as a transcription factor to regulate the expression of SWS2Aβ. And we obtained the optimal final concentration of T3 at 75 nM through preliminary experiments, and the luciferase activity of SWS2Aβ was significantly increased by the addition of T3. We speculate that even if endogenous T3 is present, its concentration may not be high enough to regulate the expression of SWS2Aβ via TRαA. Combined with the above results, we suggest that T3 may directly or indirectly regulate the expression of SWS2Aβ through TRαA.
Point 3: · Line 51, Line 52: In Japanese Flounder, do photoreceptors "secrete" opsin photopigments?
Response 3:This sentence is corrected to ‘The rod synthesizes Rhodopsin (RH1), which is responsible for vision in low light, and the cone synthesizes red-sensitive opsin (M/LWS).’
Point 4: · Lines 78-81: This description is about the mouse retina. It's better to state it clearly.
Response 4:This sentence is corrected to ‘In mice, TH signaling can promote the expression of M-cone opsin and suppress the expression of S-cone opsin through TR and play a positive regulatory role in promoting the patterned distribution of dorsal-ventral opsins.’
Point 5: · In Figure 1, what is the P. olivaceus "SWS2A(2)" ?
Response 5:We carefully revised Figure 1 and clearly labeled SWS2Aα and SWS2Aβ.

This manuscript is a resubmission of an earlier submission. The following is a list of the peer review reports and author responses from that submission.
Round 1
Author Response
Thank you for your patient review! Please see the attachment, and the corresponding modified parts have been marked in the article.
We previously neglected the duplication of opsin genes in the flounder. Ten opsin genes were found in Japanese flounders. including three blue opsin genes and four green opsin genes (RH1, LWS, SWS1, SWS2Aα, SWS2Aβ, SWS2B, RH2A-1, RH2A-2, RH2B-1, RH2B-2). In addition, the opsin genes SWS2A and RH2 of flounder adapt to the changes of light environment by regulating the development of flounder. The expression level of SWS2Aβ increases most significantly in the process of transformation, while the expression level of RH2B-1 decreases gradually while maintaining the original green sensitive function in the process of transformation (Zhengrui Z et al, 2022), which is consistent with the transformation from upper water to water in the process of transformation. Referring to the relevant literature and combining our previous transcriptome data with BLAST results on the NCBI website, we focused on five opsin genes of flounder (RH1, LWS, SWS1, SWS2Aβ, RH2B-1) in this paper. However, we have analyzed the evolutionary pattern of all ten opsin genes of flounder and have conducted an in-depth study for the five opsin genes mentioned above, and the gene names and corresponding gene IDs have been indicated in the article.

Reviewer 2 Report
In the current manuscript, Shi et al. examined the expression patterns of five retinal opsins during the metamorphosis of Japanese flounder. The authors specifically focused on the roles of the thyroid hormone signaling in opsin expression. The significant findings in the manuscript are 1) the thyroid hormone (T3) prematurely induces the expression of red, blue, and RH opsins, 2) the inhibition of thyroid hormone signaling (by thiourea) upregulates the green and UV opsins and downregulates the red and blue opsins at the end of the Climax stage (41dph), 3) cone opsins in the thiourea-treated larvae are plastic regarding the T3 response during the Climax stage (35-41dph).
The scope of this study is important, and the data looks meaningful. However, there are multiple ambiguous descriptions in the main text, which spoils the value of this manuscript.
Below are just some examples:
- There are a lot of mistakes in the reference section. The authors must correct them carefully.
- Lines 40 – 43: This manuscript has no data regarding the expression of thyroid hormone receptors. Also, this sentence contradicts the description in lines 376 – 379.
- Lines 227 – 228: This description is not consistent with the data in Figure 3D. The authors should explain why Figure 3D data differs from the NC group data in Figure 4D.
- Lines 295-296: The result of red opsin is shown in Figure 6D (not Figure 6A). In addition, there is no statistical result in Figure 6D. In the first place, Figure 6 shows the results at 12h or 24h after the T4 injection?
- Lines 299 – 301: There is no data about this description in the manuscript. If the authors are talking about Figure 6B, there is no statistical result.
- Lines 361 – 364: There is no data about this description in the manuscript.
- Lines 166 – 167: The promoter sequences of the five opsin genes of flounder “were” obtained ….
- Lines 162 – 164: This sentence is grammatically inaccurate.
- Lines 169 – 171: This sentence is grammatically inaccurate.
Other points
- Please show the T4 levels and the expression patterns of thyroid hormone receptors in each group in Figure 4A.
- The Y-axis scale in Figure 4B-F is different than that in Figure 5B-F. The authors need to explain the reason.
- There is no control in Figure 6. The authors need to show the positive and negative controls in Figure 6 (e.g., T3-induced genes, T3-suppressed genes).
- It would be better to show pictures in Figure 5A. Pictures are more convincing than cartoons.
- Lines 116: The authors need to explain how to select the 46 amino acid sequence.
- Line 120: What is the “similarity percentage”? Is this the Bootstrap value? Please provide a more detailed explanation.
- Lines 184-185: In which experiment was the two-way ANOVA used?
- Lines 376 – 379: Why do the authors specifically focus on the TRaA? The result of luciferase assays suggests that all TRs could regulate blue opsin expression. Besides, both TRa and TRb are expressed during metamorphosis. Please describe the rationale.
Author Response
Thank you for your patient review! Please see the attachment, and the corresponding modifications have been marked in the article.
Point 1: Lines 40-43: This manuscript has no data regarding the expression of thyroid hormone receptors. Also, this sentence contradicts the description in lines 376 - 379.
Response 1: I have revised this part of the content to make the results uniform, and added the thyroid hormone receptor expression data in the article.
Point 2: Lines 227-228: This description is not consistent with the data in Figure 3D. The authors should explain why Figure 3D data differs from the NC group data in Figure 4D.
Response 2: We corrected the mistake and recreated the Figure 4.
Point 3: Lines 295-296: The result of red opsin is shown in Figure 6D (not Figure 6A). In addition, there is no statistical result in Figure 6D. In the first place, Figure 6 shows the results at 12h or 24h after the T4 injection?
Lines 299-301: There is no data about this description in the manuscript. If the authors are talking about Figure 6B, there is no statistical result.
Response 3: I have made a thorough revision of the above two issues based on Figure 6. And reworked the title of Figure 6: “Relative levels of five opsin genes in adult flounder eyes after 24h injected with TH and TU.”.
Point 4: Lines 361-364: There is no data about this description in the manuscript.
Response 4: I have added relevant data to the article.
Point 5: Lines 166-167: The promoter sequences of the five opsin genes of flounder “were” obtained ….
Response 5: This sentence is corrected to “The promoter sequences of the five opsin genes of flounder were obtained from the NCBI database and used to design specific amplification primers.”.
Point 6: Lines 162-164: This sentence is grammatically inaccurate.
Response 6: Please provide your response for Point 2. (in red) Alibaba online database was used to analyze potential transcription factor binding sites (http://gene-regulation.com/pub/programs/alibaba2/index.html).
Point 7: Lines 169-171: This sentence is grammatically inaccurate.
Response 7: This sentence is corrected to "In addition, recombinant plasmids (p3×Flag-TRαA, p3×Flag-TRαB and p3×Flag-TRβ) were previously constructed and preserved in the laboratory.".
Point 8: Please show the T4 levels and the expression patterns of thyroid hormone receptors in each group in Figure 4A.
Response 8: We re-supplemented the expression patterns of thyroid hormone receptors. However, changes in T4 levels throughout the fish during metamorphosis in flounder larvae have already been published (Jie Yu. Analysis of thyroid hormone receptor mediating thyroid hormone regulation of metamorphosis in Japanese flounder (Paralichthys olivaceus) [D]. Shanghai Ocean University, 2018). And in the present study, we injected T4 into eye tissue in vivo and found no changes in the expression of relevant genes, while exogenous T3 caused blue opsin to be upregulated, which would suggest that T3 exerts a regulatory effect on the opsin gene. Moreover, it has been confirmed in most previous studies that T4 exerts a gene regulatory role by generating T3 in response to deiodinase, and its own regulatory role is very small. Therefore, in this study, the level of T4 in ocular tissues was not analyzed.
Point 9: The Y-axis scale in Figure 4B-F is different than that in Figure 5B-F. The authors need to explain the reason.
Response 9: The Y-axis scale in Figure 4 and Figure 5 is different because they normalize genes differently. Figure 4 Normalized genes are Blue opsin NC group data, and Figure 5 normalized genes are 36 dph TU group data of each opsin gene.
Point 10: There is no control in Figure 6. The authors need to show the positive and negative controls in Figure 6 (e.g., T3-induced genes, T3-suppressed genes).
Response 10: No genes promoted or repressed by T3 in dental flounder have been identified as positive and negative controls in the current relevant studies. In this part, we used uninjected flounder as the control. Our previous experiments found that the addition of T3 and TU in cultured water can affect the changes of opsin levels of flounder, so we directly injected T3, T4 and PTU (inhibit the conversion of T4 to T3) into the tissues around the eyes of flounder here to observe the changes of opsin levels.
Point 11: It would be better to show pictures in Figure 5A. Pictures are more convincing than cartoons.
Response 11: We have reworked Figure 5A in order to hopefully show the results more clearly.
Point 12: Lines 116: The authors need to explain how to select the 46 amino acid sequence.
Response 12: Based on the amino acid sequences of the five opsins, BLAST searches for similar species (mainly Osteichthyes) were conducted from the NCBI website, and common mammals, reptiles, and amphibians were selected together to construct an evolutionary tree, resulting in the screening of 46 amino acid sequences.
Point 13 : Line 120: What is the “similarity percentage”? Is this the Bootstrap value? Please provide a more detailed explanation.
Response 13: Yes, it is the Bootstrap value. This sentence is corrected to “The reliability of the phylogenetic tree was assessed by a bootstrap test with 1000 replicates.”
Point 14: Lines 184-185: In which experiment was the two-way ANOVA used?
Response 14: In this experiment, we used only a one-way ANOVA, which we corrected in the article.
Point 15: Lines 376-379: Why do the authors specifically focus on the TRaA? The result of luciferase assays suggests that all TRs could regulate blue opsin expression. Besides, both TRa and TRb are expressed during metamorphosis. Please describe the rationale.
Response 15: The results of luciferase analysis showed that all TRs could regulate the expression of SWS2Aβ, but we found that the difference in luciferase activity was not significant in the pro-SWS2Aβ + TRαA only transfected group, and the difference was significant after the addition of T3, while the luciferase activity was significantly higher in the pro-SWS2Aβ + TRαB/TRβ group, so we focused on the pro-SWS2Aβ + TRαA group.

Reviewer 3 Report
Yaxin Shi et al
Thyroid Hormone Signaling is Required for Dynamic Varia-2 tion of Opsins in the Retina during metamorphosis of the Japanese Flounder (Paralichthys olivaceus)
General comments:
The paper of Shi et all describes the shifts in opsin expression with development in a commercial fish, the Japanese flounder. As they develop from larvae to adults, many fish species undergo a transformation in the wavelengths and types of opsins, whether rod or cone, that are expressed. The paper further examines the ability of the thyroid hormones T3 and T4, and a substance that inhibits thyroid hormone synthesis, phenylthiourea (PTU), on the developmental alterations in opsin expression. The results are consistent with thyroid hormone involvement. The paper then delves into the alterations in the expression of thyroid hormone receptors, finding importance in that area as well, although the reviewer found the experimental flow and methods here were not fully explained making the data, if quite significant, difficult to interpret. The reviewer was concerned that the authors did not deal with ‘gene duplication’, an event that gives fish not just the four UV, blue, mid (Rh2) and red cone opsins, but multiple versions of blue, mid and red opsins. The reviewer found another flounder species with gene duplications and wondered why the authors hadn’t commented on this. Half of vertebrate species are fish, and not surprisingly, there were other references that might have been included. Altogether, while overall the paper was quite interesting, data rich, and clear, there were numerous faults in English grammar.
Specific comments:
Line 51: ‘The majority of vertebrate species contain four PR types: rods, short single cones, long single cones, and double cones.’ The majority of teleost species…’ Cite:
"Engstrom K. Cone types and cone arrangements in teleost retinae. Acta Zool 44: 179-243, 1963."
Line 77: ‘flounder has three types of TRαA, TRαB and TRβ’ flounder has three types of TR (TRαA, TRαB and TRβ)…
Line 93: ‘The 16 dph…’ define dph
Line 95: ‘0.1 mg/L TH’ Line 153: ‘T3 group (50.0 ppm)’, Line 176: ‘T3 at a final concentration of 75 nM’. Please use consistent units when reporting concentration, preferably in ‘nm’.
Line 99: ‘decapitated and executed’ euthanized by decapitation
Line 100: ‘larvae sample’ larvae samples
Line 141: Opsin primer table. In many teleosts including other flounder species, blue, green, and red opsins are gene duplicated is this not true of Japanese flounder? For example:
“Kasagi S, Mizusawa K, Murakami N, Andoh T, Furufuji S, Kawamura S, Takahashi A. Molecular and functional characterization of opsins in barfin flounder (Verasper moseri). Gene. 2015 Feb 10;556(2):182-91.”
Line 238: Figure 3. This is a nice figure.
Line 244: ‘To confirm whether TH regulate opsin genes expression’ To confirm whether TH regulates opsin gene expression
Line 245: ‘in the eyes tissue’ in the eye tissue
Line 246: ‘exogenous TH increased immediately T3 levels’ were you applying T4?
Line 265: wonderful figure 4 comparing thiourea and T3 effects on QPCR of opsins
Line 272: ‘For investigate further whether metamorphosis of the TU-treated flounder could be’ To investigate…
Line 284: Fig. 5. The meaning of the X-axis labels ’41 dphTU+NC and 41dphTU+TH’ needs explanation in the figure legend, particularly NC. The ‘+’ probably means ‘followed by’ whatever symbol is used here, it needs to be defined.
Line 286: ‘(A) Rescue group perverted larvae pattern diagram.’ What is the meaning of perverted in this context? The word commonly means ‘sexual deviant’. How is this a diagram? Do you mean altered behavioral pattern?
Line 294: ‘At 12 hours after T3 injection, the expression of red opsin gene was significantly increased (P<0.05) (Figure 6A).’
Figure 6A is UV opsin, and it is not increased by T3. The legend does not say whether the figure is for 12 or 24 hours post injection.
Line 296 ‘At 12 and 24 hours after T3 injection, the level of blue opsin gene was significantly increased (P<0.05), while there was no significant difference in the expression of the other three opsin genes (Figure 6C). In addition, the expression of blue opsin gene was significantly decreased (P<0.05) at 12 hours after TU injection, while the expression of the other four opsin genes was not significantly different.’ Blue opsin is shown in Fig. 6B. TU does not significantly reduce blue opsin expression in Fig. 6B.
Line 310: Rho-pro+TRs group. Please define what this is. Rho is a GTPase family. Or are you reporting on activity of the rhodopsin promoter? Since it is a dual luciferase, what is the normalizing gene for the ratio? In Fig. 7, ‘pro-Rho+TRalpha A’ how is the thyroid receptor (TR) added. Is this a gain-of-function transgenic? The methodology in the luciferase section needs fuller explanation.
Line 333: ‘Related studies have shown that the diversity of the types and expression levels of fish opsin genes are closely related to the light environment of the water body in which they live [37, 38]. One could add a Cichlid reference:
“Carleton K. Cichlid fish visual systems: mechanisms of spectral tuning. Integrative Zoology 4: 75-86, 2009.”
Line 343: ‘During metamorphosis in Paralichthys olivaceus, TH accelerate its metamorphosis process [25], but TU inhibits this process [28, 41].’ TH accelerates the metamorphosis
Line 360: ‘Similarly, TH is important for the differentiation of L-cone opposed to UV cone fate in zebrafish [44].’ TR is important…as opposed to UV-cone fate. There are two other pertinent references here:
“Deveau C, Jiao X, Suzuki SC, Krishnakumar A, Yoshimatsu T, Hejtmancik JF, and Nelson RF. Thyroid hormone receptor beta mutations alter photoreceptor development and function in Danio rerio (zebrafish). PLoS genetics 16: e1008869, 2020.”
“Nelson RF, Balraj A, Suresh T, Elias LJ, Yoshimatsu T, and Patterson SS. The developmental progression of eight opsin spectral signals recorded from the zebrafish retinal cone layer is altered by the timing and cell type expression of thyroxin receptor β2 (trβ2) gain-of-function transgenes. Eneuro 2022.”
This is a pertinent citation for TH actions on zebrafish opsin expression:
“Mackin RD, Frey RA, Gutierrez C, Farre AA, Kawamura S, Mitchell DM, and Stenkamp DL. Endocrine regulation of multichromatic color vision. Proceedings of the National Academy of Sciences 116: 16882-16891, 2019.”
Line 369: ‘A decrease in TRalpha2 in zebrafish results in an increase in the number of vertebrae expressing UV opsin’ in the number of cones expressing UV opsin
Line 376: ‘And it is hypothesized that T3 may directly regulate the expression of blue opsin promoter through TRalphaA’ It would be more careful to say ‘directly or indirectly regulate’
Author Response
Thank you for your patient review! Please see the attachment, and the corresponding modifications have been marked in the article.
Point 1: Line 51: ‘The majority of vertebrate species contain four PR types: rods, short single cones, long single cones, and double cones.’ The majority of teleost species…’ Cite: "Engstrom K. Cone types and cone arrangements in teleost retinae. Acta Zool 44: 179-243, 1963."
Response 1: We have modified the content of the foreword and used this article as a reference.
Point 2: Line 77: ‘flounder has three types of TRαA, TRαB and TRβ’ flounder has three types of TR (TRαA, TRαB and TRβ)…
Response 2: We have revised the preamble and removed this sentence.
Point 3: Line 93: ‘The 16 dph…’ define dph
Response 3: This sentence is corrected to “The 16 days post-hatching (dph) larvae...”.
Point 4: Line 95: ‘0.1 mg/L TH’ Line 153: ‘T3 group (50.0 ppm)’, Line 176: ‘T3 at a final concentration of 75 nM’. Please use consistent units when reporting concentration, preferably in ‘nm’.
Response 4: "0.1 mg/L T3" is the amount of T3 added to the culture water and "final concentration of 75 nM T3" is the amount of T3 added to the cell culture medium, but "T3 group (0.50 ppm)" is the periocular tissue injected concentration of the solution we could not determine its final concentration and it is consistent with other injected solutions T4 (0.50 ppm) and TU (50.0 ppm) units.
Point 5: Line 99: ‘decapitated and executed’ euthanized by decapitation.
Line 100: ‘larvae sample’ larvae samples
Response 5: These two mistakes have been corrected in the article.
Point 6: Line 141: Opsin primer table. In many teleosts including other flounder species, blue, green, and red opsins are gene duplicated is this not true of Japanese flounder? For example:
“Kasagi S, Mizusawa K, Murakami N, Andoh T, Furufuji S, Kawamura S, Takahashi A. Molecular and functional characterization of opsins in barfin flounder (Verasper moseri). Gene. 2015 Feb 10;556(2):182-91.”
Response 6: Ten opsin genes were found in Japanese flounders. including three blue opsin genes and four green opsin genes (RH1, LWS, SWS1, SWS2Aα, SWS2Aβ, SWS2B, RH2A-1, RH2A-2, RH2B-1, RH2B-2). In addition, the opsin genes SWS2A and RH2 of flounder adapt to the changes of light environment by regulating the development of flounder. The expression level of SWS2aβ increases most significantly in the process of transformation, while the expression level of RH2B-1 decreases gradually while maintaining the original green sensitive function in the process of transformation, which is consistent with the transformation from upper water to water in the process of transformation. Referring to the relevant literature and combining our previous transcriptome data with BLAST results on the NCBI website, we focused on five opsin genes of flounder (RH1, LWS, SWS1, SWS2Aβ, RH2B-1) in this paper.
Point 7: Line 244: ‘To confirm whether TH regulate opsin genes expression’ To confirm whether TH regulates opsin gene expression
Line 245: ‘in the eyes tissue’ in the eye tissue
Line 246: ‘exogenous TH increased immediately T3 levels’ were you applying T4?
Response 7: We removed ‘To confirm whether TH regulates opsin genes expression ’and corrected a grammatical error in the article. In addition, the phrase "exogenous TH increased immediately T3 levels" was revised to "exogenous T3 immediately increased T3 levels".
Point 8: Line 272: ‘For investigate further whether metamorphosis of the TU-treated flounder could be’ To investigate…
Response 8: This sentence is corrected to “To investigate further whether metamorphosis of the TU-treated flounder could be ...”.
Point 9: Line 284: Fig. 5. The meaning of the X-axis labels ’41 dph TU+NC and 41dphTU+TH’ needs explanation in the figure legend, particularly NC. The ‘+’ probably means ‘followed by’ whatever symbol is used here, it needs to be defined.
Response 9: We have added the following to the figure legend: “The 41 dph TU+NC and 41 dph TU+TH groups indicate that the 36 dph TU-treated larvae were transferred to natural seawater and seawater supplemented with 0.1 mg/L T3 to continue rearing until 41 dph, respectively.”.
Point 10: Line 286: ‘(A) Rescue group perverted larvae pattern diagram.’ What is the meaning of perverted in this context? The word commonly means ‘sexual deviant’. How is this a diagram? Do you mean altered behavioral pattern?
Response 10: During metamorphosis, flounder transitioned from the upper water to the bottom water. This figure shows that the metamorphosis of flounder treated with 36 dph TU was inhibited, so they remained in the upper water column, while flounder in the rescue group metamorphosed successfully and transitioned to benthic life. In addition, this sentence is corrected to “(A) Metamorphosis pattern of flounder larvae in the rescue group.”.
Point 11: Line 294: ‘At 12 hours after T3 injection, the expression of red opsin gene was significantly increased (P<0.05) (Figure 6A).’ Figure 6A is UV opsin, and it is not increased by T3. The legend does not say whether the figure is for 12 or 24 hours post injection.
Line 296 ‘At 12 and 24 hours after T3 injection, the level of blue opsin gene was significantly increased (P<0.05), while there was no significant difference in the expression of the other three opsin genes (Figure 6C). In addition, the expression of blue opsin gene was significantly decreased (P<0.05) at 12 hours after TU injection, while the expression of the other four opsin genes was not significantly different.’ Blue opsin is shown in Fig. 6B. TU does not significantly reduce blue opsin expression in Fig. 6B.
Response 11: I have completely modified the above two problems in the paper according to Figure 6. The title of Figure 6 was reworked: "Relative levels of five opsin genes in adult flounder eyes after 24h injected with TH and PTU". And the legend was added: “The Ctrl indicates the uninjected flounder, T3 indicates the injection of 0.50 ppm T3, T4 indicates the injection of 0.50 ppm T4, and PTU indicates the injection of 50.0 ppm PTU.”.
Point 12: Line 310: Rho-pro+TRs group. Please define what this is. Rho is a GTPase family. Or are you reporting on activity of the rhodopsin promoter? Since it is a dual luciferase, what is the normalizing gene for the ratio? In Fig. 7, ‘pro-Rho+TRalpha A’ how is the thyroid receptor (TR) added. Is this a gain-of-function transgenic? The methodology in the luciferase section needs fuller explanation.
Response 12: We redefined the pro-RH1 + TRs group, described the normalized genes and provided a more comprehensive interpretation of the luciferase fraction approach. This sentence is corrected to “Five recombinant plasmids of the opsin promoter region were co-transfected with p3×Flag-TRs (p3×Flag-TRαA, p3×Flag-TRαB and p3×Flag-TRβ) recombinant plasmids, respectively, normalization with data from each promoter activity group. And the results showed that the luciferase activity was significantly higher in the pro-RH1 + TRs group (RH1 promoter region plasmid was co-transfected with p3×Flag-TRs, respectively) and the group with the simultaneous addition of T3 (P<0.05) (Figure 7A).”.
Point 13: Line 343: ‘During metamorphosis in Paralichthys olivaceus, TH accelerate its metamorphosis process [25], but TU inhibits this process [28, 41].’ TH accelerates the metamorphosis
Response 13: This sentence is corrected to “During metamorphosis in Paralichthys olivaceus, TH accelerate its metamorphosis process...”.
Point 14: Line 369: ‘A decrease in TRalpha2 in zebrafish results in an increase in the number of vertebrae expressing UV opsin’ in the number of cones expressing UV opsin
Response 14: This sentence is corrected to “a decrease in TRβ2 in zebrafish leads to an increase in the number of cones expressing UV photoreceptors”.
Point 15: Line 376: ‘And it is hypothesized that T3 may directly regulate the expression of blue opsin promoter through TRalphaA’ It would be more careful to say ‘directly or indirectly regulate’
Response 15: This sentence is corrected to “and may even directly or indirectly regulate the expression of SWS2Aβ through TRαA”.
